# Leveraging machine learning to evaluate factors influencing vitamin D insufficiency in SLE patients: A case study from southern Bangladesh

**Mrinal Saha**[1], **Aparna Deb**[1], **Imtiaz Sultan**[1], **Sujat Paul**[1], **Jishan Ahmed**[2], **Goutam Saha**[3] *

1 Department of Medicine, Chattogram Medical College, Chattogram, Bangladesh, 2 Department of Mathematics, University of Barisal, Barisal, Bangladesh, 3 Department of Mathematics, University of Dhaka, Dhaka, Bangladesh

* gsahamath@du.ac.bd

**Data Availability Statement:** All relevant data are available on Gitlab: https://gitlab.com/Jishan/sle-data/-/tree/main/Data.

## Abstract

Vitamin D insufficiency appears to be prevalent in SLE patients. Multiple factors potentially contribute to lower vitamin D levels, including limited sun exposure, the use of sunscreen, darker skin complexion, aging, obesity, specific medical conditions, and certain medications. The study aims to assess the risk factors associated with low vitamin D levels in SLE patients in the southern part of Bangladesh, a region noted for a high prevalence of SLE. The research additionally investigates the possible correlation between vitamin D and the SLEDAI score, seeking to understand the potential benefits of vitamin D in enhancing disease outcomes for SLE patients. The study incorporates a dataset consisting of 50 patients from the southern part of Bangladesh and evaluates their clinical and demographic data. An initial exploratory data analysis is conducted to gain insights into the data, which includes calculating means and standard deviations, performing correlation analysis, and generating heat maps. Relevant inferential statistical tests, such as the Student's t-test, are also employed. In the machine learning part of the analysis, this study utilizes supervised learning algorithms, specifically Linear Regression (LR) and Random Forest (RF). To optimize the hyperparameters of the RF model and mitigate the risk of overfitting given the small dataset, a 3-Fold cross-validation strategy is implemented. The study also calculates bootstrapped confidence intervals to provide robust uncertainty estimates and further validate the approach. A comprehensive feature importance analysis is carried out using RF feature importance, permutation-based feature importance, and SHAP values. The LR model yields an RMSE of 4.83 (CI: 2.70, 6.76) and MAE of 3.86 (CI: 2.06, 5.86), whereas the RF model achieves better results, with an RMSE of 2.98 (CI: 2.16, 3.76) and MAE of 2.68 (CI: 1.83, 3.52). Both models identify Hb, CRP, ESR, and age as significant contributors to vitamin D level predictions. Despite the lack of a significant association between SLEDAI and vitamin D in the statistical analysis, the machine learning models suggest a potential nonlinear dependency of vitamin D on SLEDAI. These findings highlight the importance of these factors in managing vitamin D levels in SLE patients. The study concludes that there is a

**Funding:** The authors received no specific funding for this work.

**Competing interests:** The authors have declared that no competing interests exist.

high prevalence of vitamin D insufficiency in SLE patients. Although a direct linear correlation between the SLEDAI score and vitamin D levels is not observed, machine learning models suggest the possibility of a nonlinear relationship. Furthermore, factors such as Hb, CRP, ESR, and age are identified as more significant in predicting vitamin D levels. Thus, the study suggests that monitoring these factors may be advantageous in managing vitamin D levels in SLE patients. Given the immunological nature of SLE, the potential role of vitamin D in SLE disease activity could be substantial. Therefore, it underscores the need for further large-scale studies to corroborate this hypothesis.

## Introduction

SLE is characterized by inflammation, pain, and damage to organs and tissues. Symptoms of SLE can vary greatly from person to person and can include fatigue, joint pain, and stiffness, skin rash (typically on the face or scalp), fever, hair loss, mouth ulcers, anemia, swelling in the legs, sun sensitivity, chest pain, enlarged lymph nodes. SLE can also affect internal organs and can lead to serious complications if left untreated. Treatment for SLE is tailored to each individual patient based on the severity of their symptoms and the organs affected. The relationship between vitamin D and SLE was first described in 1995, and it appears that vitamin D receptors are expressed on immune cells and may play an important role in regulating the immune system [1]. Moreover, individuals with SLE often have low levels of vitamin D, and introduce of vitamin D supplements may improve the health condition of SLE patients [2].

SLE is a prolonged autoimmune illness that can affect numerous organ systems in the body. The exact reason for SLE is not fully understood, but it is believed to be a combination of genetic, ecological, and hormonal issues. In recent years, it is seen that SLE patients with low vitamin D have a poorer prognosis. Several research has been conducted to investigate the association between vitamin D and SLEDAI, but the results have been inconsistent, with some studies showing a correlation and others finding no association. It is to be noted that vitamin D exists in two major forms: vitamin D2 and vitamin D3 and both are biologically inactive and need to be hydroxylated in the liver and kidney to make them active known as calcitriol (1,25-dihydroxyvitamin D3). Calcitriol binds to vitamin D receptors in the body and then modulates gene expression and cell differentiation [2].

Low vitamin D is a widespread public health issue and about 1 billion people globally are affected by that. The incidence of lower vitamin D varies widely depending on factors such as geographic location, skin pigmentation, age, and lifestyle. In 2007, a study reported that 80% of South Asian people were vitamin D insufficient, with 40% having severely low vitamin D level [3]. Studies have shown that the pervasiveness of vitamin D insufficiency in North India ranges between 30–90%. It is found that 91% and 84% of young girls, and pregnant women were vitamin D insufficient. Low vitamin D in the North Indian population is also due to the high pigmentation of the skin. And such skin faces difficulty to create vitamin D from sunlight [4].

Bangladesh is a tropical country located at 24 degrees north latitude, which means that it receives abundant sunlight year-round. However, despite the abundance of sunlight in Bangladesh, several research found that the occurrence of low vitamin D is excessive among Bangladeshi women and it was almost 83%. Such high occurrence can be attributed to a number of factors, such as cultural practices that restrict sun exposure, particularly for women, poor nutrition, and reduced vitamin D intake in the diet. The prevalence of vitamin D insufficiency is particularly high among low-income lactating women [5]. Islam & Amin [6] found that 38%

of high-income women and 50% of low-income women had hypovitaminosis D (vitamin D levels <15.5 ng/mL) Furthermore, the study found that about 39% of Bangladeshi young university girls and 30% of veiled women had low vitamin D levels. These findings suggest that cultural practices and poor nutrition are major contributing factors to lower vitamin D in women in Bangladesh. Some important research related to vitamin D and SLE patients is presented in Table 1.

## Related research work

This research addresses two prominent issues. The primary motivation is the high prevalence of vitamin D insufficiency in SLE patients, especially in the southern part of Bangladesh. Understanding the contributing factors to this condition and its potential impact on disease activity is crucial for patient care. In this study, while we utilize established algorithms, Linear Regression, and Random Forest, the novelty lies in their application to the specific problem of vitamin D insufficiency in SLE patients and the combination of these algorithms with an extensive feature importance analysis. Traditionally, studies investigating vitamin D insufficiency in SLE patients have relied on conventional statistical methods. However, our approach leverages the power of machine learning to uncover potentially intricate patterns and relationships in the data that could not be detected by standard techniques. For instance, we performed an in-depth feature importance analysis using the Random Forest's intrinsic feature importance, permutation-based feature importance, and SHAP values, which helps to understand the relative contribution of each feature to the model's predictions. This is a significant advancement over traditional statistical methods and can provide much richer insights. By integrating these machine learning models with traditional statistical analyses, our research aims to provide a comprehensive understanding of the dynamics influencing vitamin D levels in SLE patients. This innovative approach addresses the limitations of previous methods and enhances our understanding of vitamin D insufficiency in SLE patients. The findings from this study can potentially contribute to better clinical management strategies for SLE patients and drive further investigations in this field.

## Materials and methods

### Study sites, study design, and duration

Chattogram, also known as Chittagong, is the second-largest city in Bangladesh with a population of around 7.62 million and a total area of 5282.92 square km as shown in Fig 1. The city is home to 22 government hospitals, with Chattogram Medical College Hospital (CMCH) being one of the largest and most well-equipped. A case-control study was conducted at CMCH from December 1, 2017 to December 31, 2018, during which both qualitative and quantitative data were collected.

### Ethical approval, sample size, inclusion, and exclusion criteria and sampling

Written permission is collected from the Research Technical Committee, Ethical Committee, and Peer Review Committee of Chattogram Medical College in order to conduct this research. The research reference number and date of approval are provided as M/PG/2017/366 with date 27/12/2017. The study recruited a patient group of 50 patients who came to the hospital during the study period, using a specified case definition as described by the American College of Rheumatology (ACR, 1997) criteria for SLE classification and all patients who gave written consent to participate in the study and duration of participation is Dec 2017 to Dec 2018.

**Table 1. Details of participants, cut-off value for vitamin D, number of SLE patients.**

| *Ref.* | Participants, mean age (years), duration of SLE (years) | Participants Country & Classification criteria | Cut-off levels for 25(OH)D (*ng/ml*) Insufficiency (I), Deficiency (D) & measurement process | Number of Patients with Vitamin D levels |
|---|---|---|---|---|
| Present study | SLE: 50, Age: 25.26±9.80, duration: Less than 6 months (36 patients), More than a year (14 patients) | Bangladeshi male & female, ACR | D<10; 10≤I<30, Normal (N)≥30, EIM | *SLE group*: Insufficient: 48 |
| [7] | SLE: 123, Controls: 240 | African Americans & Caucasians, ACR | D<10; 10≤I<30, Normal (N)≥30 | *SLE group*: Deficient: 22 Insufficient: 101 |
| [8] | SLE: 92, Age: 41, duration: 7 | Spanish male and female, ACR | D<10; 10≤I<30, Normal (N)≥30, CLIA | *SLE group*: Deficient: 14 Insufficient: 69 |
| [9] | SLE: 36, Age: 29.8, Controls: 26, Age: 32.8 | Brazilian women, ACR | D<20; 20≤I<30, Normal (N)≥30, RIA | - |
| [10] | SLE: 181, Age: 43.2±10.6, duration: 11.9±8.6 | White, African American, Hispanic & Asian women, ACR | D<10; 10≤I<30, Normal (N)≥30 | *SLE group*: Deficient: 36 Insufficient: 113 |
| [11] | SLE: 198 | Asian, African American, Caucasians, and Hispanic, ACR | D<10; 10≤I<30, Normal (N)≥30 | - |
| [12] | SLE: 80, Age: 43±14, duration: 9.5 | Spanish male and female, ACR | D<10; 10≤I<30, Normal (N)≥30 | *SLE group*: Deficient: 5 Insufficient: 83 |
| [13] | SLE: 40, Age: 25.3±4.2 | Iranian women, ACR | Severe<5, 5≤Moderate<10, 10≤Mild<16, RM | *SLE group*: Mild: 5 Moderate: 25 Severe: 7 |
| [14] | SLE: 50, Age: 29.38±9.2, duration:5.43±4.56, Controls: 30, Age: 30.4±7.1 | Egyptian female, ARA | D<10; 10≤I<30, Normal (N)≥30, Rkit | *SLE group*: Deficient: 18 Insufficient: 32 *Control group*: Deficient: 5 Insufficient: 17 |
| [15] | SLE: 104, Age: 36.21 ± 10.21, Controls: 49, Age: 35.33 ± 6.18 | Korean male and female, ACR | D<10; 10≤I<30, Normal (N)≥30, IA | *SLE group*: Insufficient: 17 *Control group*: Insufficient: 2 |
| [16] | SLE: 177, Age: 44.9 ± 14.6 | Hungarian people, ACR | D<15; 15≤I<30, Normal (N)≥30, CLIA | - |
| [17] | SLE: 78, Age: 36.9±10.67, Controls: 64, Age: 38.13±9.704 | Brazilian male and female, ACR | I<30, Normal (N)≥30 | *SLE group*: Insufficient: 45 |
| [18] | SLE: 195, Age: 42.6±13.9, duration: 9.2, Controls: 201 | Brazilian male and female, ACR | Severe<10, 10≤D<20; 20≤I<30, Normal (N)≥30, ,C | *SLE group*: Severe: 11 Deficient: 52 Insufficient:63 |
| [19] | SLE: 73, Age: 47.3±13.7, duration: 3.5 | Spanish women, ACR | D<10; 10≤I<30, Normal (N)≥30 | *SLE group*: Insufficient: 50 |
| [20] | SLE: 40, Age: 29.75±6.93, duration:5.23±4.21, Controls: 20, Age (M,F): 27.90±5.88, 29.90±6.85 | Egyptian male & female, ACR | D<12; 12≤I<30, Normal (N)≥30, IEIkits | *SLE group*: Deficient: 27 Insufficient: 7 *Control group*: Deficient: 6 Insufficient: 6 |
| [21] | SLE: 129, Age: 28.14±8.43, duration: 2.90±2.66, Controls: 100, Age: 31.18±5.32 | Indian male and female. ACR | D<10; 10≤I<30, Normal (N)≥30, ELISA | - |

*(Continued)*

**Table 1.** (Continued)

| Ref. | Participants, mean age (years), duration of SLE (years) | Participants Country & Classification criteria | Cut-off levels for 25(OH)D (*ng/ml*) Insufficiency (I), Deficiency (D) & measurement process | Number of Patients with Vitamin D levels |
|---|---|---|---|---|
| [22] | SLE: 58, Age: 39.78, Controls: 58, Age: 44.81 | Bahraini male and female, ACR | D<12; 12≤I<20, Optimal≥20, CLIA | *SLE group*: Deficient: 26 Insufficient: 25 *Control group*: Deficient: 17 Insufficient: 29 |
| [23] | SLE: 45, Age: 28.8±7.9, duration: 11.3±9.8, Controls: 40, Age: 30.4±9.6 | Egyptian male and female, SLICC | D<10; 10≤I<30, Normal (N)≥30, ELISA | *SLE group*: Deficient: 21 Insufficient: 24 *Control group*: Insufficient: 40 |
| [24] | SLE: 137, Age: 45.9±11.6, Duration:7.7±3.4, Active:31, Inactive:106 | Mexican women, ACR | D<10; 10≤I<30, Normal (N)≥30, ACLMI | *SLE group*: Deficient: 4 Insufficient: 122 |
| [25] | SLE: 90, Age: group A: 33.78±6.2, group B: 35.69±6.8, duration: group A: 9.53±3.8, group B: 10.98±3.5 | Iranian people, ACR | I<30, Normal (N)≥30 | - |
| [26] | SLE: 81, Age: G1 (n = 21): 36.4±7.6, G2 (n = 30): 35.2±8.7, G3 (n = 30): 37.7±8.9 | Saudi-Arabian people | D<20; 20≤I<30, Normal (N)≥30, Liaison IXT | *SLE group*: Deficient: 35 Insufficient: 32 |
| [27] | SLE: 35, Age: 12.9±3.12, Active: 35, Inactive: 35 | Taiwan children | D<20; 20≤I<30, Normal (N)≥30, AEA | *SLE group*: *Inactive* Insufficient: 33 *Active* Deficient: 28 |
| [28] | SLE: 100, Age: 29.8 ± 9.8, duration:31 | Paraguayan male & female, SLICC | D<20; 20≤I<30, Normal (N)≥30, CMIA | - |
| [29] | SLE: 50, Age: 28±13.34 | Bangladeshi male and female, ACR | D<20; 20≤I<30, Normal (N)≥30, ELISA | *SLE group*: Deficient: 37 Insufficient: 7 |

The small sample size is considered to be a result of time constraints and limited funding resources, which made it difficult to conduct a larger study. The study had established a set of exclusion criteria to ensure that the results obtained were as accurate and reliable as possible. Patients with certain pre-existing conditions such as End-Stage Renal Disease, Diabetes Mellitus, severe sepsis, SLE with overlap, osteoporosis, osteomalacia, first-degree relatives of SLE or any connective tissue disease, and individuals with psychiatric conditions were excluded from the study as these conditions could potentially affect the results and skew the findings. Furthermore, bed-ridden patients, patients receiving bisphosphonates, pregnant women and lactating mothers, and patients who did not give their consent were also excluded from the study to minimize any potential risks and ensure the safety and well-being of the participants.

## Laboratory data

The study performed CBC, ESR, CRP, ANA, anti-dsDNA, vitamin D level or 25(OH)D, 24-hour urinary total protein, and renal Lupus nephritis patients. The method used to measure the inactive form of vitamin D3, 25(OH)D status, was the electrochemiluminescence immunoassay method in the serum of SLE patients. Four milliliters of venous blood were obtained from each patient maintaining eight hours of fasting. Also, blood, urine, and tissue samples were collected from SLE patients for clinical investigation and the duration of the collection is Dec 2017 to Dec 2018. We also had access to these patients' information until 3 July 2020.

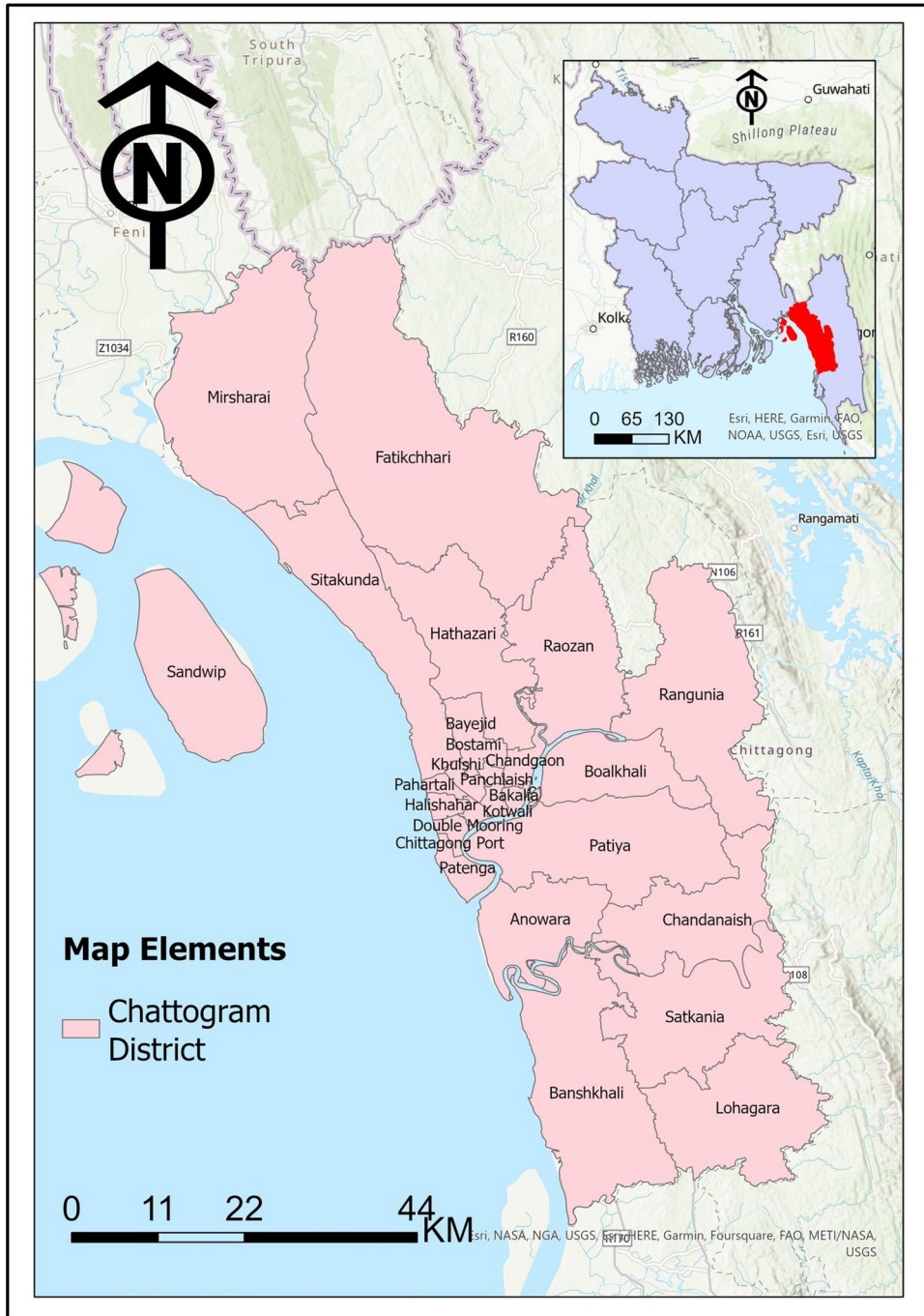

**Fig 1. Region map of Chattogram.**

### Data collection, collation, and analysis

The study collected qualitative data from the patients through face-to-face interviews in the Medicine, Rheumatology, Nephrology, and Dermatology wards of the CMC hospital. A pre-tested questionnaire was used for this study, and all demographic data, such as age, gender, educational status, geographic area, and monthly family income were recorded. Clinical data,

including drugs, duration of sun exposure (hours/day), and SLEDAI were also recorded. Laboratory testing data of the patients were also collected from the sample testing laboratory of the hospital. The following link provides free access to the data. https://gitlab.com/Jishan/sle-data/-/tree/main/Data.

## Summary statistics

In this study, the vitamin D levels of SLE patients were described in relation to their clinical presentations, using mean and standard deviation. Laboratory-reported data for vitamin D were categorized according to the following classification: normal (N) levels being $\geq$ 30 ng/ml, deficient (D) levels being < 10 ng/ml, and insufficient (I) levels falling between 10 and 30 ng/ml [7, 10, 14, 21, 23, 24]. The vitamin D levels were then assessed against various demographic characteristics of the SLE patients. All statistical analyses were conducted using the open-source software JASP (Link: https://jasp-stats.org/). By examining the distribution of vitamin D levels across different demographic and clinical factors, this study offers valuable insights into the health concerns faced by underprivileged and marginalized SLE patients in Bangladesh. This information, in conjunction with the findings from the Random Forest regression model and SHAP values, can contribute to a better understanding of the factors that impact vitamin D levels among SLE patients and inform targeted interventions to address their unique health needs.

## Machine learning (ML) approach

Traditional regression analysis relies on assumptions such as linearity and additivity of the relationship between response and explanatory variables, statistical independence, homoscedasticity (constant variance), and normality of the errors [30]. Moreover, the presence of non-linearities and interactions can make it difficult to design efficient regression models. In this study, machine learning approaches are employed to understand the relationship between vitamin D and various factors related to SLE disease. Random forests (RF) regression model, a widely used ML approach, is capable of handling multicollinearity, feature interactions, and non-linearities [31]. RF regression method, inspired by the high performance of ML approaches, uses bootstrap training data and randomness of the explanatory variables to generate an uncorrelated forest of decision trees. Importantly, the RF model is robust to overfitting and computationally less expensive. The number of trees in the forest and the size of the feature subset to consider while searching for the best split are two crucial hyperparameters of the RF model [32].

In this study, the optimal hyperparameters are obtained using a three-fold Cross Validation (CV) approach. A linear regression (LR) model is also included as a baseline model for comparison. Models are evaluated using the root mean squared error (RMSE) and mean absolute error (MAE) regression evaluation metrics. Both RMSE and MAE scores are reported with 95% confidence intervals (CI) [33] by leveraging the bootstrap resampling technique. The test data is utilized to perform the resampling technique with replacement for 50,000 times. The 95% CI interval is constructed on the [2.5, 97.5] percentile boundaries. Impurity-based feature importance, permutation-based feature importance, Shapley values-based SHAP feature importance, and partial dependence plots are also incorporated in this study to understand the individual feature contributions to vitamin D levels [34].

Permutation importance is a reliable feature importance measure, as it is model-agnostic and takes into account the feature interaction effects. In contrast, SHAP feature importance provides a more comprehensive explanation of an RF regression model's output compared to impurity-based and permutation-based feature importance approaches. SHAP assigns a

contribution value to each feature for each prediction made by the RF regression model, allowing for an examination of how each feature individually contributes to the model's output and how the contribution of each feature depends on the values of the other features [34]. SHAP has been found to be a more effective approach for explaining predictions compared to traditional linear regression effect sizes, particularly for more complex models such as the RF regression model. This is because SHAP considers the interaction between features and can provide more accurate and comprehensive explanations of the model's output. In this study, SHAP waterfall plots are presented to understand the influence of each feature on the predictions using Shapley values. By utilizing machine learning approaches, specifically the Random Forest regression model and SHAP feature importance, this study seeks to uncover a deeper understanding of the factors that contribute to low vitamin D levels in SLE patients. This comprehensive approach provides valuable insights into the complex relationships between vitamin D and various aspects of SLE disease, ultimately contributing to a better understanding of the health concerns facing underprivileged and marginalized SLE patients in Bangladesh.

## Experimental setup and model configuration

ML computations are performed on a multi-core machine with CPU: 11th Gen Intel Core @ 2.30GHz (8 cores), RAM: 32 GB RAM, and OS: Windows 11 Home.

The following Python implementations are presented in this paper:

- LR model implemented by the sci-kit learn [35] module in Python

- RF model implemented by the sci-kit learn module [35] in Python

To implement ML models, the entire dataset is split into two parts, 80% data for training and 20% for testing. *GridSearchCV from the* sci-kit learn module [35] is used to optimize the hyperparameters of the RF regression model using 3-fold CV. The values of the complexity parameter, the maximum depth of the tree, and the number of trees in the forest are found as 0.00001, 2, and 50 respectively as best hyperparameters in *GridSearchCV* for the RF model.

In this study, overfitting was mitigated through several strategies:

Cross-validation: A 3-Fold cross-validation strategy was employed, which helps to ensure that the model generalizes well to unseen data. In cross-validation, the dataset is divided into 'k' subsets or 'folds'. The model is then trained on k-1 folds and tested on the remaining fold. This process is repeated k times, with a different fold used as the testing set each time. This approach provides a robust estimate of the model's predictive performance.

Hyperparameter tuning: Optimal values for the model's hyperparameters were selected through a grid search approach, which finds the combination of hyperparameters that result in the best cross-validation performance.

Feature importance analysis: Through feature importance analysis methods like permutation feature importance and SHAP values, irrelevant features, or 'noise', were identified. Models can overfit if they learn patterns from noise, so this analysis helped to ensure that the models focused on the genuine, informative features.

Ensemble method: The Random Forest algorithm was used, which is an ensemble of decision trees. By averaging the predictions of multiple decision trees, the Random Forest model effectively reduces the risk of overfitting.

Lastly, we validated our approach by computing bootstrapped confidence intervals. This provided robust uncertainty estimates that further ensured the prevention of overfitting.

These steps helped to ensure that our model had good predictive performance, not just on the training data, but also on unseen data.

**Table 2. Strengths and weaknesses of different methods.**

| Methods | Strengths | Weaknesses |
|---|---|---|
| Correlation and t-tests (Previous Methods) | Simple to understand and interpret, Widely accepted in the scientific community | May not detect complex, nonlinear relationships, Can not handle high-dimensional data well, Cannot easily provide insights into the relative importance of features |
| Linear Regression (Our Method) | Can model linear relationships, Provides coefficients that can be interpreted as the strength of the relationship between each feature and the outcome | Assumes a linear relationship, Can be sensitive to outliers |
| Random Forest (Our Method) | Can model nonlinear and complex relationships, Robust to outliers, Provides intrinsic feature importance | Requires careful tuning to avoid overfitting, Interpretability can be more challenging than linear methods |

A comparison table that highlights the strengths and weaknesses of our method compared to previous methods is included as Table 2.

The Table 2 illustrates that while traditional methods like correlation and t-tests are simple and widely accepted, they might not capture complex relationships in the data. On the other hand, our proposed methods, Linear Regression and Random Forest, are capable of modeling such complex relationships and providing insights into the relative importance of features. However, they also come with their own challenges, such as the need for careful tuning and the interpretability of the results. We believe that the combined use of traditional statistical tests and machine learning models in our study provides a more comprehensive understanding of the factors influencing vitamin D levels in SLE patients.

## Results

In this study, 84% and 16% of the patients were female and male respectively. Also, 52%, 20%, and 10% of the patients were treated with high-dose Prednisolone, high-dose Prednisolone with methylprednisolone, low-dose Prednisolone respectively. And only 10% were not taking any steroidal medications. The study assigned values of 1 for male, 0 for female, 0 for body mass index (BMI) level 24 or less, 1 for BMI level 25 or over, 1 for sun exposure more than one hour, 0 for sun exposure less than one hour, 1 for positive renal involvement, and 0 for negative renal involvement. Table 3 provides detailed descriptive data.

The average age of participants is 25.26 years, with a standard deviation of 9.80 years, indicating a moderate spread around the mean. The youngest participant is 13 years old, while the oldest is 50 years old. The sex distribution shows a higher proportion of females in the sample. Participants have an average hemoglobin (HB) level of 10.12 g/dL, with a standard deviation of 1.03 g/dL, and their erythrocyte sedimentation rate (ESR) has a mean value of 61.68 mm/h and

**Table 3. Descriptive statistics.**

| Variables | Age | Sex | HB | ESR | CRP | FSS | SLEDAI | BMI | Sun | RI | Vit D |
|---|---|---|---|---|---|---|---|---|---|---|---|
| Mean | 25.26 | 0.16 | 10.12 | 61.68 | 11.71 | 5.50 | 17.26 | 0.20 | 0.60 | 0.46 | 19.56 |
| Median | 22.00 | 0.00 | 10.15 | 67.00 | 3.99 | 5.80 | 17.00 | 0.00 | 1.00 | 0.00 | 18.00 |
| SD | 9.80 | 0.37 | 1.03 | 30.11 | 17.11 | 1.03 | 13.31 | 0.40 | 0.50 | 0.50 | 5.34 |
| Skewness | 1.10 | 1.91 | -0.80 | -0.13 | 2.55 | -0.80 | 0.58 | 1.55 | -0.42 | 0.17 | 0.36 |
| Kurtosis | 0.20 | 1.73 | 0.08 | -0.95 | 7.39 | 0.08 | -0.53 | 0.41 | -1.90 | -2.06 | -0.89 |
| Maximum | 50.00 | 1.0 | 16.60 | 118.0 | 86.00 | 7.00 | 47.00 | 1.00 | 1.00 | 1.00 | 30.30 |
| Minimum | 13.00 | 0.0 | 5.90 | 4.00 | 0.00 | 2.90 | 1.00 | 0.00 | 0.00 | 0.00 | 10.50 |

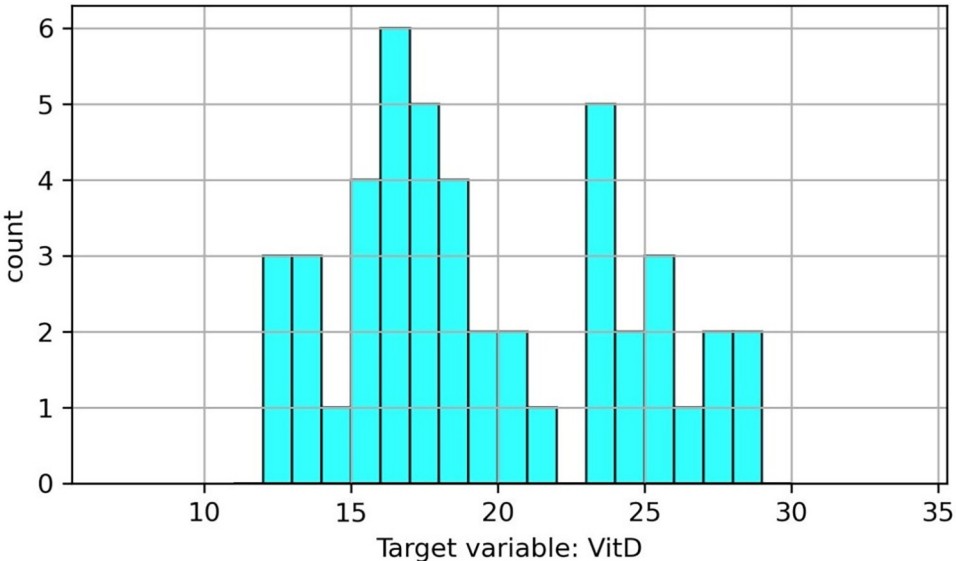

**Fig 2. Distribution of target variable (vitamin D levels).**

a relatively large standard deviation of 30.11 mm/h. The C-reactive protein (CRP) levels show an average value of 11.71 mg/L and a high standard deviation of 17.11 mg/L, suggesting a wide range of values. The Fatigue Severity Scale (FSS) scores have a mean value of 5.50 and a standard deviation of 1.03. The Systemic Lupus Erythematosus Disease Activity Index (SLEDAI) scores exhibit an average value of 17.26 with a standard deviation of 13.31. Finally, the mean body mass index (BMI) is 0.20, with a standard deviation of 0.40, and the average vitamin D level is 19.56, with a standard deviation of 5.34. Overall, these descriptive statistics provide insights into the distribution and central tendencies of the data, which can be valuable for further analyses.

Fig 2 illustrates that the distribution of the response variable, Vitamin D, does not conform to a normal distribution. Nevertheless, it is still possible to model it using the linear regression (LR) model, as linear regression analysis does not necessitate normality for either the explanatory variables or the target variable. Moreover, a non-parametric random forest (RF) model does not require stringent assumptions.

Adopting an exploratory approach serves as an excellent initial step in identifying potential relationships between variables. Nonparametric methods, such as Spearman's rank-order correlation (Fig 3), are useful for analyzing nonlinear relationships. It is observed that some variables, including Age, Sex, BMI, Sun, RI, FSS, and CRP, exhibit an insignificant correlation with Vitamin D, indicating little or no association between these variables and Vitamin D levels. However, a positive correlation exists between hemoglobin (Hb) and Vitamin D, suggesting that individuals with higher Vitamin D levels also tend to have elevated hemoglobin levels. This may be attributed to various factors, such as the role of Vitamin D in regulating iron metabolism, which is crucial for hemoglobin production. Furthermore, a strong negative relationship is observed between SLEDAI and Vitamin D levels, indicating that individuals with higher SLEDAI scores typically have lower Vitamin D levels.

A heat map serves as an effective tool for visualizing the relationship between two variables. The heat map depicted in Fig 4 demonstrates the connection between SLEDAI and Vitamin D. Patients with very high SLEDAI scores (ranging from 20 to 30) tend to exhibit Vitamin D insufficiency, while patients with low or mild SLEDAI scores (ranging from 1 to 5) generally

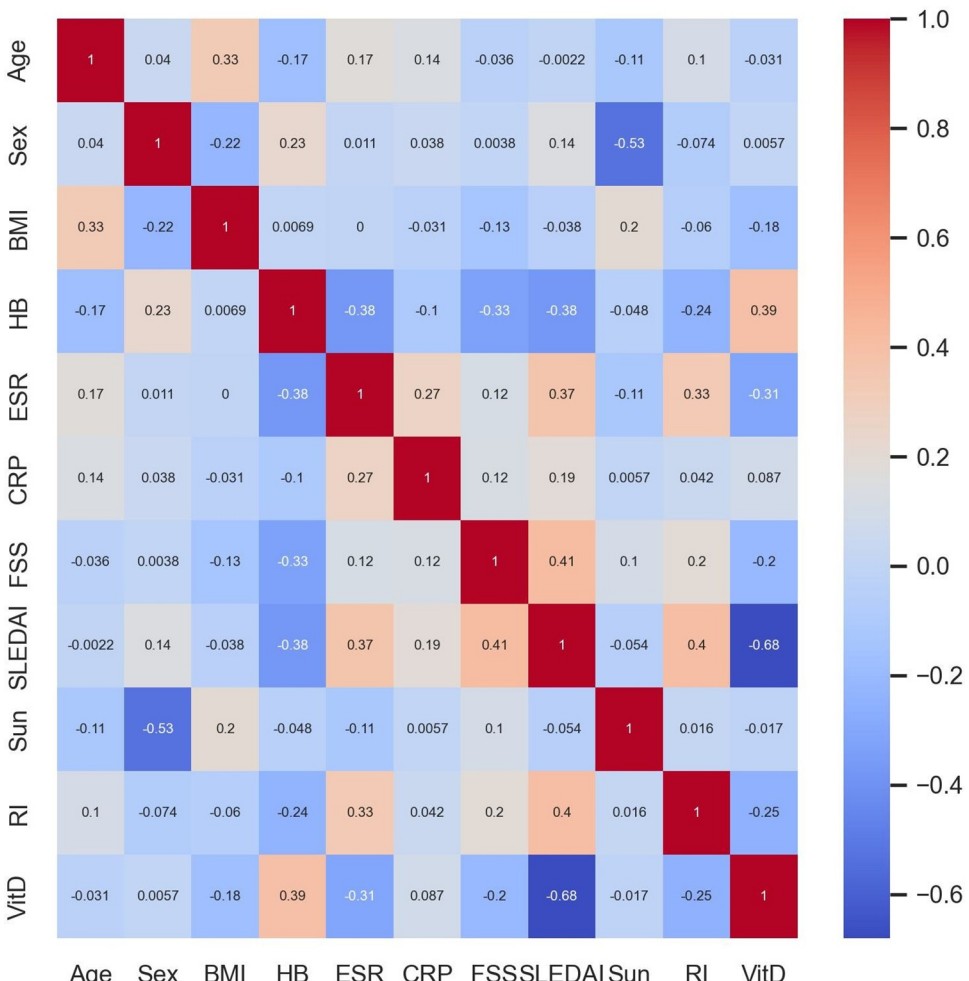

**Fig 3. Spearman's correlation map between the variables.**

have Vitamin D levels close to normal. This relationship aligns with the findings of Begum et al. [29], suggesting that patients with low SLEDAI scores are in the recovery stage and maintain normal Vitamin D levels.

Table 4 was obtained by conducting independent two-sample t-tests to compare the mean vitamin D levels among SLE patients across different groups based on gender, sun protection usage, and sun exposure duration. The t-tests were performed to assess if there are any significant differences in the means between the groups. The resulting p-values, along with 95% confidence intervals for the differences in means, were reported to determine the presence of significant associations between the factors and vitamin D levels. These findings were then interpreted in conjunction with the correlation and heat map analyses to gain a comprehensive understanding of the relationships between the variables under study.

In Table 4, an analysis of the mean vitamin D levels in SLE patients is presented, considering factors such as gender, sun protection usage, and sun exposure duration. The findings indicate no significant association between gender and vitamin D levels (p = 0.7056), which contrasts with the results of Yan et al. [36], who reported a substantial effect of gender on vitamin D levels. This discrepancy may be attributed to differences in the study populations, methodologies, or sample sizes, and further investigation is warranted. In addition, the results show

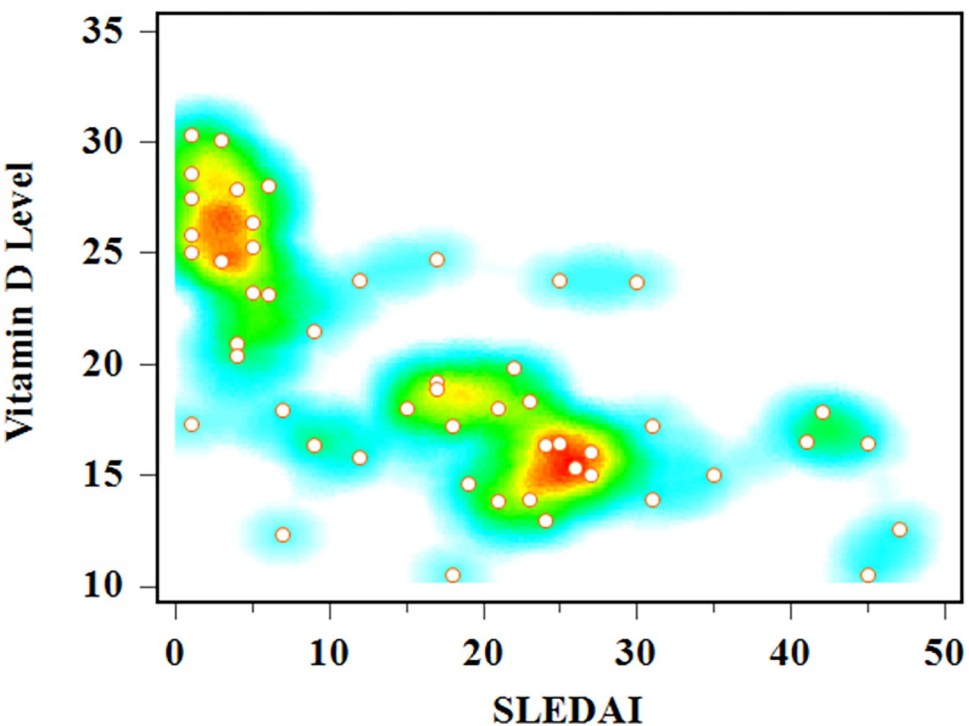

**Fig 4. Heat map presentation of vitamin D level & SLEDAI.**

no significant relationship between the use of sun protection measures (such as sunscreen or umbrellas) and vitamin D levels in SLE patients (p = 0.6119). This finding is consistent with the observations from the heat map and correlation analysis, where factors such as age, sex, BMI, sun exposure, and sun protection measures exhibited weak or no correlation with vitamin D levels. However, there is a significant relationship (p = 0.0089) between vitamin D levels and the duration of sun exposure, where SLE patients who receive more than an hour of sun exposure have higher vitamin D levels compared to those with less sun exposure. This finding suggests that longer sun exposure may improve vitamin D levels in SLE patients, despite recommendations for photosensitive SLE patients to avoid sun exposure [37]. Magro et al. [38] also mentioned low vitamin D levels were common due to avoiding sun exposure and the use of sun protective measures. The heat map and correlation analysis further support this observation, as they reveal a positive association between sun exposure and vitamin D levels, particularly in individuals with low SLEDAI scores who are in the recovery stage.

**Table 4. Vitamin D level of SLE patients considering gender, residence, sun protection, and sun-exposure.**

| Variables | Type of participant (n = 50) | 25 (OH) D: Mean ± SD | p-value | 95% CI |
|---|---|---|---|---|
| Gender | | | | |
| Male | SLE patients | 20.23 ± 6.29 | 0.7056 | (-3.39, 4.97) |
| Female | | 19.44 ± 5.22 | | |
| Sun Protection | | | | |
| Sunscreen or Umbrella | SLE patients | 19.22 ± 6.23 | 0.6119 | (-2.53, 4.25) |
| No use | | 20.08 ± 5.17 | | |
| Sun Exposure | | | | |
| < 1 hour | SLE patients | 18.5 ± 4.7 | 0.0089** | (1.28, 8.46) |
| > 1 hour | | 23.37 ± 5.56 | | |

**Table 5. Performance of ML models on test data.**

| Model | RMSE | MAE |
|---|---|---|
| LR | 4.83 [2.70, 6.76] | 3.86 [2.06, 5.86] |
| RF | **2.98 [2.16, 3.76]** | **2.68 [1.83,3.52]** |

To further substantiate our findings, we employed machine learning models and compared the performance of the Random Forest (RF) model with the baseline Linear Regression (LR) model using bootstrapped test data (Table 5).

In order to validate the findings from the descriptive statistics, t-test, and correlation analysis, advanced machine learning models were employed, comparing the performance of the Random Forest (RF) model with the baseline Linear Regression (LR) model using bootstrapped test data (Table 5). As demonstrated through the previously mentioned descriptive statistics, t-test, and correlation results, variables such as Hb, SLEDAI, and sun exposure display significant relationships with vitamin D levels, while other variables like age, sex, BMI, sun protection measures, and CRP exhibit weak or negligible correlation. These insights offer valuable information regarding the potential predictors of vitamin D levels in SLE patients and can be further investigated using more sophisticated modeling techniques, such as the Random Forest (RF) and Linear Regression (LR) models. Building upon the insights obtained from the descriptive statistics, t-test, and correlation analysis, the Random Forest (RF) model was employed to further investigate the relationships between the variables and the vitamin D levels in SLE patients.

The performance of the RF model was compared to the LR model using bootstrapped test data, and the outcomes were reported in Table 5 based on the Root Mean Squared Error (RMSE) and Mean Absolute Error (MAE) measures. Table 5 reveals that the RF model outperformed the LR model in terms of both RMSE and MAE scores. Specifically, the RF model achieved the lowest RMSE score of 2.98 while maintaining a narrow CI, and the RF model also produced the lowest MAE score of 2.68 with a narrow CI. The superior performance of the RF model is likely due to its ability to capture complex, nonlinear relationships between the predictors and the target variable, as evidenced by the earlier findings from the descriptive statistics, t-test, and correlation analysis. However, it is essential to note that feature importance in the RF model is sensitive to cardinality. To investigate potential issues with cardinality further, a non-informative variable with random numbers is also added to compute the feature importance. This approach helps ensure a more robust and reliable assessment of the variable importance and better understanding of their contribution to the RF model's predictive performance.

The findings from the RF model's feature importance analysis, as shown in Fig 5A and 5B, are consistent with the earlier results from Table 3 (descriptive statistics), Table 4 (t-test results), and Fig 3 (Spearman correlation analysis). In these previous analyses, it was identified that SLEDAI, Hb, CRP, ESR, and age exhibit significant relationships with vitamin D levels. The RF model's feature importance analysis reinforces these findings, as it also reveals that these five features are the most contributing factors towards the predictions of vitamin D levels, even in the presence of additional non-informative features.

Figs 5B, 6, and 7 further support these conclusions, as the contributions of BMI, FSS, Sun, RI, and gender are no better than the random noise. The five most contributing features (SLEDAI, Hb, CRP, ESR, and age) remain consistent across various feature importance approaches, including permutation-based feature importance and Shapley values-based feature importance.

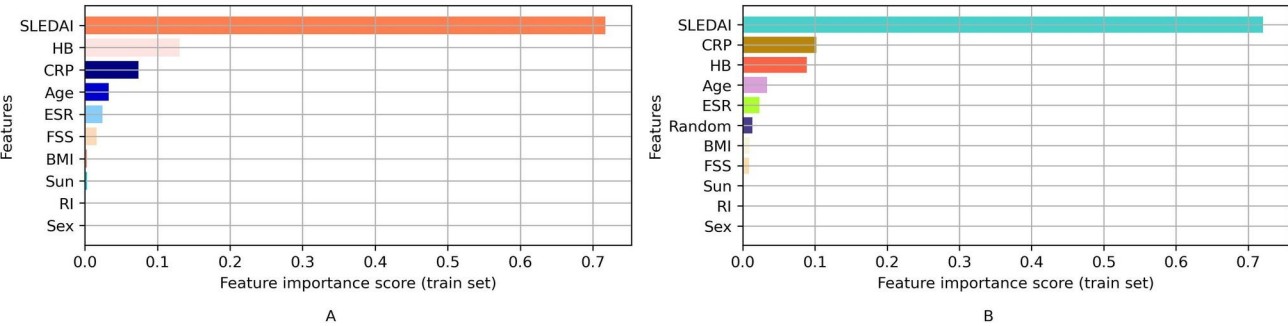

**Fig 5.** Variable importance of the RF regression model on the training data: A. variable importance without non-informative feature, B. variable importance with non-informative feature.

The local explanation of the vitamin D level predictions in Fig 7 aligns with the feature importance results and shows how individual feature contributions are aggregated to form the final predictions. The single-variable partial dependence (PDP) plots in Fig 8 also confirm the importance of these features, as they demonstrate the relationship between the individual features and the predicted outcome of the RF regression model. The PDP plots are consistent with the variable importance, permutation importance, and SHAP importance measures, further corroborating the findings from Tables 3, 4, and Fig 3.

Fig 6 displays the most influential features in descending order based on the magnitude of SHAP values across the test dataset. The blue and red dots situated to the right of zero represent a positive influence on vitamin D levels, while dots to the left signify a negative impact on the predictions. Local explanations of the predictions are provided in Fig 8, illustrating the contributions of individual features. In Fig 7, the final predictions are calculated by aggregating the positive (red) and negative (blue) contributions of each feature from the base expected value. Fig 8A demonstrates that the test observation #1's final prediction is 25.967, resulting from the accumulation of individual feature contributions with a base value of 19.569. Similarly, for test observation #7, the RF model predicts vitamin D levels of 16.362, as shown in

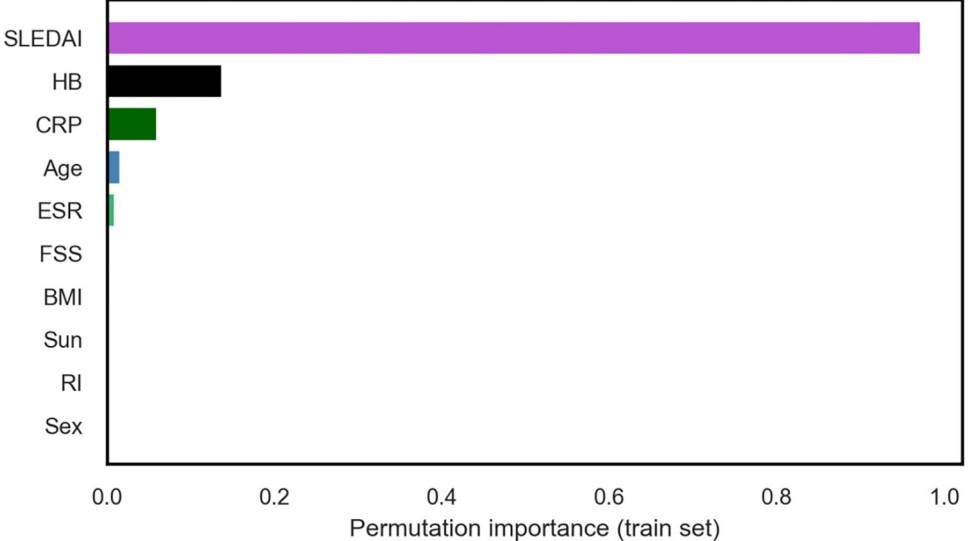

**Fig 6. Permutation-based feature importance of the RF regression model on the testing data.**

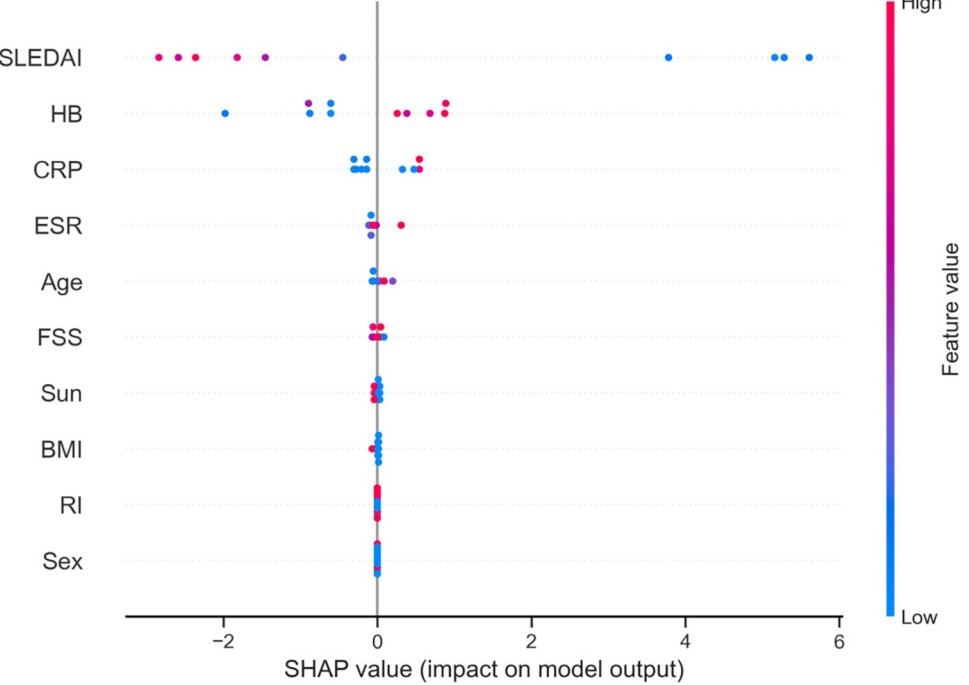

**Fig 7. Shapley values-based feature importance on the testing data.**

Fig 8A. These observations indicate that the individual feature contributions align with the feature importance plot presented in Fig 6.

This study employs single-variable partial dependence plots (PDP) to visualize the relationship between individual features and the predicted outcomes of the RF regression model. Fig 9 presents PDP plots, incorporating ten Individual Conditional Expectation (ICE) curves and a PD line. In the top-left PDP plot, the dashed orange line indicates minimal dependence of vitamin D levels on age, while the top-right plot reveals a nonlinear and monotonically increasing partial dependence of vitamin D levels on Hb values greater than 10. Similarly, a weak nonlinear partial dependence is observed for the CRP variable in the bottom-left plot. In contrast, a strong partial dependence is evident in the bottom-right plot, demonstrating that the partial dependence of vitamin D levels on SLEDAI is nonlinear and declines sharply for values exceeding 5. Consequently, the PDP plots align with the variable importance, permutation importance, and SHAP importance measures. Nonetheless, minor variations in ICE curves (light blue lines) are noticeable in all PDP plots, except for the bottom-right PDP plot. These findings highlight the importance of considering SLEDAI, Hb, CRP, ESR, and age as crucial factors in the clinical management of SLE patients and the need for further investigation into their potential therapeutic implications.

## Discussion

This research aimed to find how vitamin D level correlates with SLEDAI. No relationship association between SLEDAI and the level of vitamin D is observed using ML. This finding is similar to the findings of Kim et al. [15], Muñoz-Ortego et al. [19], Abbasi et al. [39], and Petri et al. [40]. According to Gado et al. [23], low vitamin D is common for SLE patients and observed a strong correlation between the level of vitamin D with ESR, and FSS. In the present research considering the ML approach, it is found that lower levels of vitamin D are strongly

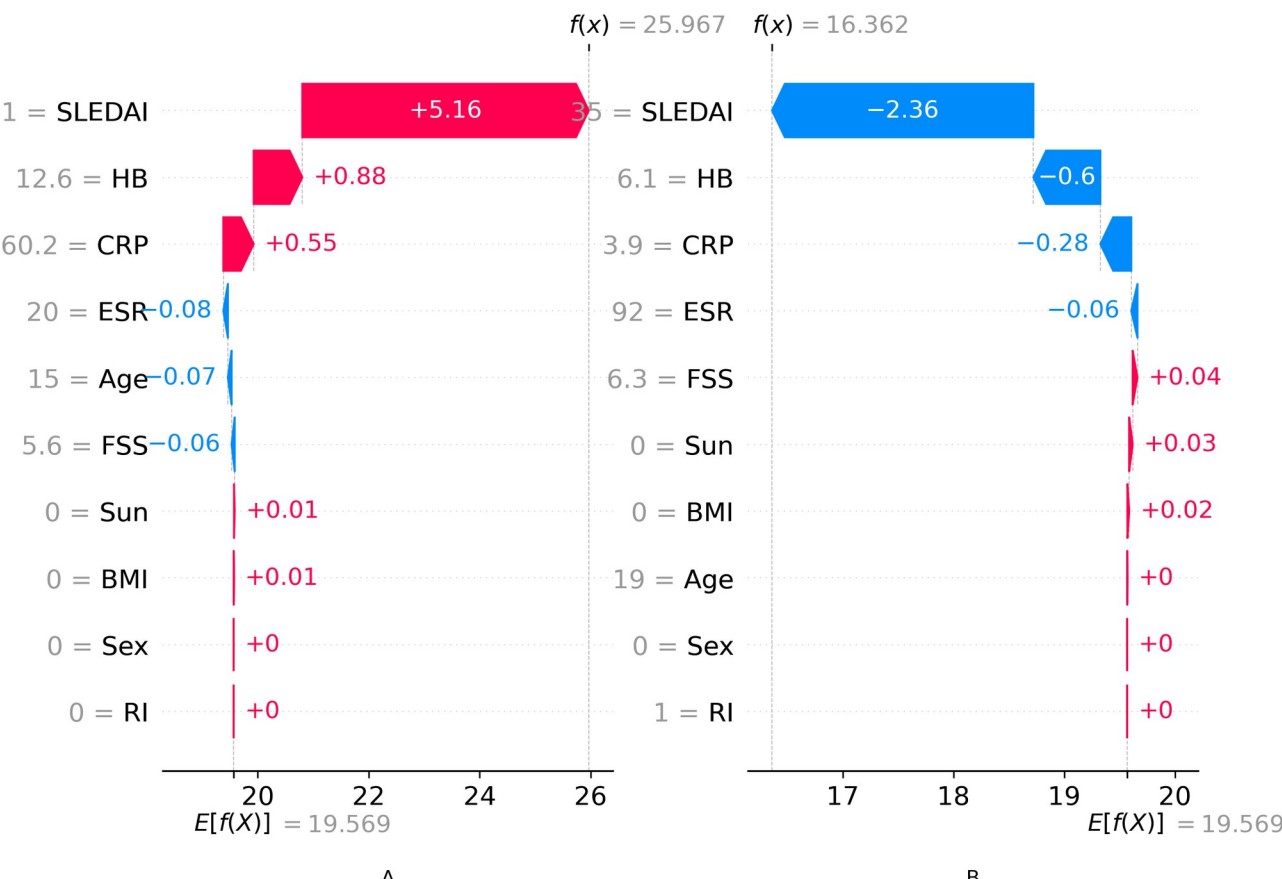

**Fig 8.** Local explanation of the Vitamin D level predictions in two test observations: A. Predicted vitamin D level in test observation #1, B. Predicted vitamin D level in test observation #7.

connected to ESR. However, no relationship between vitamin D and FSS is observed that is in line with Stockton *et al.* [41].

SLE patients have low vitamin D levels having an average level of 19.54 ± 5.29 ng/ml. This suggests that SLE patients have vitamin D insufficiency. This finding is similar to that of other researchers in case-control studies, such as those conducted by Mok and Lau [42], Mandal et al. [21], and Farid et al. [22]. It is important to note that lower levels of vitamin D are a common issue in many populations, not just in SLE patients, and it is important for all individuals to monitor and manage their vitamin D levels. The study conducted by Abaza et al. [43] on the Egyptian population found similar results, with the control group having optimal vitamin D levels of 79 ± 28.7 ng/ml, while the SLE patients were vitamin D insufficient with a mean level of 17.6 ± 6.9 ng/ml. This further supports the idea that lower levels of vitamin D may be a risk factor for poor SLE disease outcomes.

It is observed that there is a positive relationship between age and vitamin D, which is consistent with the findings of Khazaei et al. [44]. However, it's worth noting that a study conducted by Arabi et al. [45] on Lebanese people observed that older patients had lower levels of vitamin D compared to younger patients, which may refer to other factors influencing vitamin D levels in SLE patients, such as disease activity or medications. A statistically strong association between BMI and vitamin D is observed, as supported by research by El-Sherbiny et al.

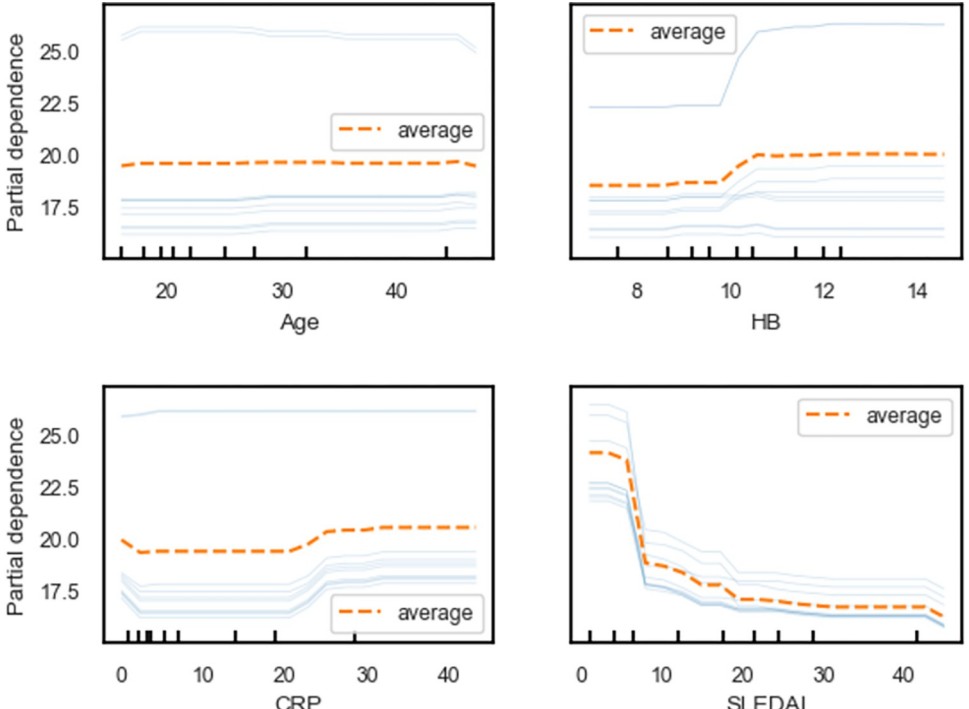

**Fig 9. Partial dependence of vaccination uptake on the influential variables for the training data.**

[46]. Similarly, studies have also found a strong association between renal involvement in SLE patients and vitamin D levels, as supported by Park et al. [47].

It is seen that lower levels of vitamin D are common in rural slum areas due to various factors such as pollution, crowding, lack of education, less sun exposure, inadequate intake of a balanced diet and vitamin D supplements, and low socioeconomic conditions as reported by Hossain et al. [48]. Additionally, a study by Karimzadeh et al. [25] found that vitamin D supplements did not significantly improve the level of vitamin D, which may be due to factors such as disease activity, poor absorption, or other underlying health conditions. These findings emphasize the importance of identifying and addressing the underlying causes of lower levels of vitamin D, as well as the need for more research to determine the most effective ways to improve the level of vitamin D in this population.

Based on the information provided in Table 6 and the references cited, it can be concluded that most SLE patients have vitamin D insufficiency. The mean value of vitamin D for SLE patients in our study is similar to the reference values found in other studies, such as those by Kamen et al. [7], Ruiz-Irastorza et al. [8], Ben-Zvi et al. [11], García-Carrasco et al. [24], and Karimzadeh et al. [25], which further supports the idea that lower levels of vitamin D are common in SLE patients.

## Conclusion

The study finds a high prevalence of vitamin D insufficiency, particularly among females, suggesting a crucial role of vitamin D in SLE patients, who are predominantly female. Most SLE patients in this study have low vitamin D levels associated with high SLEDAI scores. Moreover, the study indicates that low levels of Hb, CRP, and ESR may contribute to these low vitamin D levels. The causal relationship between vitamin D and SLE outcomes remains unclear, but

**Table 6. Variation of mean level of vitamin D.**

| Ref. | Mean 25 (OH) D levels |
|---|---|
| Present study | SLE: 19.56 ± 5.34 *ng/ml* |
| [7] | SLE: 21.6 *ng/ml*, Controls: 27.4 *ng/ml* |
| [8] | SLE: 22 ± 12 *ng/ml* |
| [10] | SLE: 27.1 ± 11.9 *ng/ml* |
| [11] | SLE: African Americans: 14.2 *ng/ml*, Hispanics: 20.5 *ng/ml*, Asians: 22 *ng/ml*, Caucasians: 29 *ng/ml* |
| [12] | SLE: 24.8 ± 10.1 *ng/ml* |
| [13] | SLE: 9.68 ± 0.84 *ng/ml* |
| [14] | SLE: 10.52 ± 2.23 *ng/ml*, Controls: 12.28 ± 1.87 *ng/ml* |
| [15] | SLE: 42.49 ± 15.08 *ng/ml*, Controls: 52.72 ± 15.19 *ng/ml* |
| [16] | SLE: 26.88 ± 13.25 *ng/ml* |
| [17] | SLE: 29.3 *ng/ml*, Controls: 33.12 *ng/ml* |
| [18] | SLE: 25.51 ± 11.43 *ng/ml* |
| [20] | SLE: 13.84 ± 12.16 *ng/ml*, Controls: 22.73 ± 11.73 *ng/ml* |
| [22] | SLE: 12.27 *ng/ml*, Controls: 15.98 *ng/ml* |
| [23] | SLE: 12 ± 2.3 *ng/ml*, Controls: 21.1 ± 3.2 *ng/ml* |
| [24] | SLE (Active): 19.3 ± 4.5 *ng/ml*, (Inactive): 19.7 ± 6.8 *ng/ml* |
| [25] | SLE: Group A: 17.36 ± 4.26 *ng/ml*, Group B: 16.78±4.39 |
| [26] | SLE: G1: 15.4±2.9 *ng/ml*, G2: 17.1±5.3 *ng/ml*, G3: 26.5±7.9 *ng/ml* |
| [27] | SLE (Active): 12.0 ± 7.2 *ng/ml*, (Inactive): 15.4 ± 7.4 *ng/ml* |
| [28] | SLE: Week 0: 31.02±11.91 *ng/ml*, Week 24: 31.59±13.89 *ng/ml* |
| [29] | SLE: 16.70 ± 8.83 *ng/ml* |

vitamin D supplementation may improve prognosis. Further studies are needed to confirm these findings, and any vitamin D supplementation should be undertaken under the guidance of a healthcare provider due to potential side effects and interactions.

## Limitations

This study does not take into account corticosteroid therapy, complement levels, anti-dsDNA antibody titers, concomitant rheumatologic therapies, and ongoing vitamin D or calcium supplementation. However, future research will be conducted to include these factors.

## Supporting information

**S1 File. STROBE checklist.**
(DOCX)

**S2 File. Questionnaire and permission letter.**
(DOCX)

## Author Contributions

**Conceptualization:** Mrinal Saha.

**Data curation:** Mrinal Saha.

**Formal analysis:** Aparna Deb, Imtiaz Sultan.

**Methodology:** Jishan Ahmed.

**Software:** Goutam Saha.

**Supervision:** Sujat Paul.

**Visualization:** Jishan Ahmed.

**Writing – original draft:** Mrinal Saha, Aparna Deb, Imtiaz Sultan, Sujat Paul, Jishan Ahmed, Goutam Saha.

**Writing – review & editing:** Jishan Ahmed, Goutam Saha.

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
