## [Decision Letter · Decision Letter 0]

13 Jun 2023

PGPH-D-23-00763

Vitamin D levels of SLE Patients: Statistical and Machine Learning Approaches

Dear Dr. Saha,

Thank you for submitting your manuscript to PLOS Global Public Health. After careful consideration, we feel that it has merit but does not fully meet PLOS Global Public Health’s publication criteria as it currently stands. Therefore, we invite you to submit a revised version of the manuscript that addresses the points raised during the review process.

Your manuscript has been evaluated by two reviewers, and their comments are appended below.

The reviewers have provided general comments about the work reported, as well as specific opportunities for improvement; please ensure you address in particular the comments from Reviewer #1 regarding the study design and the detail in reporting of the methodology.

Additionally, one or more of the reviewers has recommended that you cite specific previously published works. The works referenced may not be directly related to the submitted manuscript. As such, please note that it is not necessary or expected to cite the works requested by the reviewer, and that citation of these works is at your discretion, depending on whether you assess the works to be sufficiently relevant to this study.

We look forward to receiving your revised manuscript.

Kind regards,

Hugh Cowley

Staff Editor

Journal Requirements:

1. Please send a completed 'Competing Interests' statement, including any COIs declared by your co-authors. If you have no competing interests to declare, please state "The authors have declared that no competing interests exist". Otherwise please declare all competing interests beginning with twhe statement "I have read the journal's policy and the authors of this manuscript have the following competing interests:"

2. We have noticed that you have uploaded Supporting Information files, but you have not included a list of legends. Please add a full list of legends for your Supporting Information files after the references list. 

3. Please provide separate figure files in .tif or .eps format only and remove any figures embedded in your manuscript file. Please also ensure all files are under our size limit of 10MB.

4. Fig 1: please (a) provide a direct link to the base layer of the map (i.e., the country or region border shape) and ensure this is also included in the figure legend; and (b) provide a link to the terms of use / license information for the base layer image or shapefile. We cannot publish proprietary or copyrighted maps (e.g. Google Maps, Mapquest) and the terms of use for your map base layer must be compatible with our CC-BY 4.0 license. 

Additional Editor Comments (if provided):

Reviewers' comments:

Reviewer's Responses to Questions

**Comments to the Author**

1. Does this manuscript meet PLOS Global Public Health’s publication criteria? Is the manuscript technically sound, and do the data support the conclusions? The manuscript must describe methodologically and ethically rigorous research with conclusions that are appropriately drawn based on the data presented.

Reviewer #1: Yes

Reviewer #2: Yes

2. Has the statistical analysis been performed appropriately and rigorously?

Reviewer #1: Yes

Reviewer #2: Yes

3. Have the authors made all data underlying the findings in their manuscript fully available (please refer to the Data Availability Statement at the start of the manuscript PDF file)?

Reviewer #1: No

Reviewer #2: Yes

4. Is the manuscript presented in an intelligible fashion and written in standard English?

Reviewer #1: Yes

Reviewer #2: Yes

5. Review Comments to the Author

Reviewer #1: Dear author(s), your statistical analysis part is good with neat explanations. But coming to ML approaches, RF approach is susceptible to overfitting, so my suggestion is to try other models and make a fair comparison with those approaches. Do feature selections properly. If possible give a sample dataset in the article. Also accuracies of models should be noted and compared. You can also try a deep model like CNN to justify that your ML models are suitable for this type application over DL models or not.

Reviewer #2: 1. The title is unclear.

2. The results section of the abstract of the article does not report any results or values quantitatively.

3. The materials and methods section of the abstract is written very unintelligibly.

4. The motivation is not clear. Why was this work done? What problem does it address that previous methods could not?

5. Some state-of-the-art papers on CNN and machine learning in the field of paper should be taken into account, such as:

(https://link.springer.com/chapter/10.1007/978-3-030-98253-9_20)

(https://bmcbioinformatics.biomedcentral.com/articles/10.1186/s12859-022-04965-8)

(https://ro-journal.biomedcentral.com/articles/10.1186/s13014-021-01906-2)

(https://link.springer.com/chapter/10.1007/978-3-031-27420-6_7)

6. Add a Related Work section to the paper.

7. Include a comparison table in the related work that highlights the strengths and weaknesses of the proposed method as well as previous methods.

8. The novelty of the algorithm needs to be incorporated.

9. Describe how to deal with overfitting in your model.

10. Besides the language, the paper needs improvement.

6. PLOS authors have the option to publish the peer review history of their article (what does this mean?). If published, this will include your full peer review and any attached files.

**Do you want your identity to be public for this peer review?** For information about this choice, including consent withdrawal, please see our Privacy Policy.

Reviewer #1: **Yes: **Dr. B Padmaja

Reviewer #2: No

---

## [Decision Letter · Decision Letter 1]

18 Sep 2023

Leveraging Machine Learning to Evaluate Factors Influencing Vitamin D Insufficiency in SLE Patients: A Case Study from Southern Bangladesh

PGPH-D-23-00763R1

Dear Dr. Saha,

We are pleased to inform you that your manuscript 'Leveraging Machine Learning to Evaluate Factors Influencing Vitamin D Insufficiency in SLE Patients: A Case Study from Southern Bangladesh' has been provisionally accepted for publication in PLOS Global Public Health.

Best regards,

Julia Robinson

Executive Editor

Reviewer Comments (if any, and for reference):

Reviewer's Responses to Questions

**Comments to the Author**

1. If the authors have adequately addressed your comments raised in a previous round of review and you feel that this manuscript is now acceptable for publication, you may indicate that here to bypass the “Comments to the Author” section, enter your conflict of interest statement in the “Confidential to Editor” section, and submit your "Accept" recommendation.

Reviewer #1: All comments have been addressed

2. Does this manuscript meet PLOS Global Public Health’s publication criteria? Is the manuscript technically sound, and do the data support the conclusions? The manuscript must describe methodologically and ethically rigorous research with conclusions that are appropriately drawn based on the data presented.

Reviewer #1: Yes

3. Has the statistical analysis been performed appropriately and rigorously?

Reviewer #1: Yes

4. Have the authors made all data underlying the findings in their manuscript fully available (please refer to the Data Availability Statement at the start of the manuscript PDF file)?

Reviewer #1: No

5. Is the manuscript presented in an intelligible fashion and written in standard English?

Reviewer #1: Yes

6. Review Comments to the Author

Reviewer #1: Dear Author(s),

Kindly highlight the areas where you have done the modifications according to reviewer comments.

7. PLOS authors have the option to publish the peer review history of their article (what does this mean?). If published, this will include your full peer review and any attached files.

**Do you want your identity to be public for this peer review?** For information about this choice, including consent withdrawal, please see our Privacy Policy.

Reviewer #1: **Yes: **Dr. B Padmaja
